# A Simple and Rapid Protocol for the Isolation of Murine Bone Marrow Suitable for the Differentiation of Dendritic Cells

**DOI:** 10.3390/mps7020020

**Published:** 2024-02-27

**Authors:** Runqiu Song, Mariam Bafit, Kirsteen M. Tullett, Peck Szee Tan, Mireille H. Lahoud, Meredith O’Keeffe, Anthony W. Purcell, Asolina Braun

**Affiliations:** Department of Biochemistry and Molecular Biology and Immunity Program, Biomedicine Discovery Institute, Monash University, Clayton, VIC 3800, Australia; rson0021@student.monash.edu (R.S.); mariam.bafit@monash.edu (M.B.); kirsteen.tullett@monash.edu (K.M.T.); peck.tan@monash.edu (P.S.T.); mireille.lahoud@monash.edu (M.H.L.); meredith.okeeffe@monash.edu (M.O.); anthony.purcell@monash.edu (A.W.P.)

**Keywords:** murine bone marrow isolation, primary cell culture, bone-marrow-derived dendritic cells, conventional dendritic cells, cDC1, cDC2, plasmacytoid dendritic cells, antigen presentation, cross-presentation

## Abstract

The generation of bone-marrow-derived dendritic cells is a widely used approach in immunological research to study antigen processing and presentation, as well as T-cell activation responses. However, the initial step of isolating the bone marrow can be time-consuming, especially when larger numbers of precursor cells are required. Here, we assessed whether an accelerated bone marrow isolation method using centrifugation is suitable for the differentiation of FMS-like tyrosine kinase 3 ligand-driven dendritic cells. Compared to the conventional flushing method, the centrifugation-based isolation method resulted in a similar bone marrow cell yield on Day 0, increased cell numbers by Day 8, similar proportions of dendritic cell subsets, and consequently a higher number of type 1 conventional dendritic cells (cDC1) from the culture. Although the primary purpose of this method of optimization was to improve experimental efficiency and increase the output of cDC1s, the protocol is also compatible with the differentiation of other dendritic cell subsets such as cDC2 and plasmacytoid dendritic cells, with an improved output cell count and a consistent phenotype.

## 1. Introduction

Dendritic cells (DCs) are immune cells specialized in processing and presenting peptide antigens to lymphocytes and therefore are also referred to as professional antigen-presenting cells. They are crucial players in the detection of pathogens and the regulation of immune responses and are studied widely in an array of murine models. However, DCs comprise only 1–2% of peripheral blood mononuclear cells and isolating high numbers ex vivo is a challenge requiring the use of FMS-like tyrosine kinase 3 ligand (Flt3L)-producing tumor models or recombinant Flt3L and/or granulocyte-macrophage colony-stimulating factor cytokines to expand the available DC pool in vivo [1,2]. Alternatively, the in vitro generation of DCs from bone marrow supplied with Flt3L can yield higher numbers of cells which are classified into three main DC subsets: plasmacytoid dendritic cells (pDCs), type 1 conventional dendritic cells (cDC1s), and type 2 conventional dendritic cells (cDC2s) [3,4,5]. Whilst all of them express CD11c, these subsets can be differentiated from each other by various additional cell surface expression markers. For example, pDCs which specialize in antiviral immunity can be distinguished from cDCs by the high expression of B220 (CD45R), PDCA-1 (CD317), and Siglec-H [6]. Murine cDC1s possess a high intrinsic ability to cross-present exogenous antigens on major histocompatibility complex I (MHC I) to CD8^+^ T cells and natural killer cells, and express high levels of CD24, XCR1, and Clec9A (CD370) [7,8,9,10]. cDC2s are able to activate and regulate a range of CD4^+^ T helper cells, such as priming type 2 T helper (Th2) cells against parasitic invasion or Th17 cells against extracellular bacteria. The identifying feature of in vitro expanded murine cDC2s is their high expression of CD11b and SIRPα [6].

The cDC1 lineage is the predominant population of DCs capable of sufficiently presenting exogenous antigens to CD8^+^ T cells through the MHC I pathway, a process termed cross-presentation [7]. Due to the central role of this type of antigen presentation in CD8^+^ T-cell-mediated immune responses, there is high value in studying cross-presentation using ex vivo expanded cDC1s to improve vaccination and cancer immunotherapy design [11,12]. However, cross-presentation is an intrinsically limited process and some assays, such as the isolation of cross-presented MHC I-bound ligands from cDC1s, require not only a high cell input but also optimal cross-presentation capacity of the in vitro generated cDC1s.

Conventionally, the first step of in vitro DC differentiation requires the isolation of murine bone marrow from femurs and tibiae by flushing the bone marrow from the bone cavity with a syringe and needle [13,14,15]. However, this process can be labor-intensive and time-consuming, especially when an assay requires high input numbers of precursors isolated from multiple mice. Additionally, specific details such as the handling of bones while flushing require some practice. For example, the insertion of the needle into the bone cavity may damage some bones and therefore impact the achieved cell yield. In contrast, the centrifugation of bones requires minimal time and preparation. Although the centrifugation of bone marrow has been proposed for other assays and cell types, we are not aware of a direct comparison of the two methods for the differentiation of DCs [16,17]. Given that particularly the differentiation of cultures with a high proportion of cDC1s and high viability can require a prolonged setup period, researchers might be hesitant to leave behind well-established but laborious bone marrow flushing protocols. It is conceivable that the centrifugation of bone marrow might raise concerns about downstream effects on the differentiation potential of DC progenitors. As hematopoietic stem cells are highly regulated by mechanical properties in their environment, their gene expression and subsequent fate may be rapidly and substantially altered by the mechanical forces applied during either the flushing or the centrifugation process, which could reduce survival or differentiation capacity in primary cell culture [18,19,20]. To formally evaluate the outcomes of the isolation method on the differentiation of DC and their functional identity, we tested both methods side by side and compared the composition of Flt3L-supplied murine bone marrow cultures at the end of an 8-day differentiation protocol.

We found that the optimized centrifugation-based preparation of murine bone marrow is more time-efficient and yields equivalent numbers of DCs (cDC1, cDC2, and pDC) compared to the conventional flushing method, without changes to the phenotype of cells.

## 2. Experimental Design

### 2.1. Materials

Cautionary notes:

The osmolarity of the complete DC medium should be adjusted to 308 ± 2 osmol/L using sterile distilled water or sterile 1 M NaCl in PBS. Plasticware and FCS should be pre-tested for their use in the primary bone marrow culture since DC differentiation can be affected by the brand and lot of tissue culture flasks and FCS composition. Equally, the optimal Flt3L concentration should also be pre-tested for each batch.

Table 1, Table 2 and Table 3 list the used plasticware, cell culture reagents and reagents for flow cytometry staining respectively.

### 2.2. Equipment

Class II biological safety cabinet;Centrifuge;Water bath;Single channel pipettes;Serological pipettes;Inverted phase contrast microscope;Humidified incubator at 37 °C and 10% CO_2_;Hemocytometer;Dissection instruments, including MAYO Scissors 14.5 cm Straight Tungsten Carbide (Elite Medical, cat. 13-5200) to cut bone epiphyses.

## 3. Procedure

### 3.1. Bone Dissection (Sterile Work Required)

Day 0:Disinfect euthanised mice thoroughly with 80% ethanol and briefly dry the coat with paper towels, transfer mice into a biosafety cabinet;Cut skin from the middle of the lower abdomen along both hind legs;Remove excess muscle tissue to expose the leg bones (femur and tibia);Remove bones by dislocating the femur from the pubic bone;Remove excess muscle and tendons from the bones;Separate the femur and tibia;Transfer bones into ice-cold RF2 medium and keep them on ice until ready to proceed with the bone marrow isolation step (Section 3.2/Section 3.3).

### 3.2. Bone Marrow Isolation Using the Flushing Method

Cut open both ends of the bone to expose the bone marrow;Fill a 20 mL syringe with 20 mL ice-cold complete DC medium and attach to a 21-gauge needle;Flush out the bone marrow into a centrifuge tube (Figure 1A);Re-suspend the bone marrow aggregates by gently pipetting several times with a serological pipette.

### 3.3. Bone Marrow Isolation Using the Centrifugation Method

Use a 21-gauge needle to pierce through the bottom of a sterile 0.5 mL microcentrifuge tube and place it into a sterile 1.5 mL microcentrifuge tube with the lid removed;Cut each bone open at the epiphysis; do not discard the epiphysis. Position the 2 bones and 2 epiphyses with the cut sides downwards inside the 0.5 mL microcentrifuge tube (Figure 1B);Add 100–150 µL ice-cold DC medium into the 0.5 mL tube containing bones and close its cap;Centrifuge at 4000× *g* for 15 s at 4 °C; the bone marrow will pellet in the 1.5 mL tube. Observe visually for a color change towards a cleared white bone, and spin one more time in case the bone marrow is still remaining in the bone;Discard the bones and 0.5 mL tubes, transfer the bone marrow from the 1.5 mL tubes into a 50 mL centrifuge tube, and add approximately 5 mL DC media, keeping the tube on ice. Each 50 mL tube can contain a maximum of bone marrow from 4 bones of 1 mouse (i.e., 2 tibiae + 2 femurs);Top up to 10 mL of ice-cold DC medium after the bone marrow has been collected into the tube;Pipette gently to make a single-cell suspension.

### 3.4. Bone Marrow Cell Culture and DC Generation

For the following procedures, avoid vigorous pipetting.

Collect isolated bone marrow into a 50 mL conical centrifuge tube in DC medium and centrifuge at 400× *g* for 7 min at 4 °C;Remove supernatant and re-suspend cells in RBC lysis buffer. The volume of RBC lysis buffer depends on the number of mice: 1 mL lysis buffer for each mouse used (2 femurs, 2 tibiae);Mix by gentle pipetting and incubate for 30–40 s;Immediately top up with 9 mL ice-cold DC medium and underlay with 3–5 mL FCS.Centrifuge (400× *g* for 5 min at 4 °C) to pellet cells at the bottom. Re-suspend the bone marrow in 30 mL cold DC media;Pass cells through a 70 nm cell strainer, underlay with 3–5 mL FCS;Re-suspend cells in a defined volume of cold DC medium and count cells;Culture bone marrow cells at 1.5 × 10^6^ cells/mL/3.2 cm^2^ in DC medium + Flt3L at 200 ng/mL. Accordingly, culturing can be performed in flasks or plates under the following conditions:96-well round bottom plate: 0.3 × 10^6^ cells in 200 µL of medium per well;6-well plate: 4.5 × 10^6^ cells in 3 mL of medium per well;25 cm^2^ flask: 15 × 10^6^ cells in 10 mL medium;75 cm^2^ flask: 45 × 10^6^ cells in 30 mL medium;10 cm dish: 25.5 × 10^6^ cells in 17 mL medium.Differentiate the bone marrow culture for 8 days. During the 8-day incubation period in a humidified incubator at 37 °C and 10% CO_2_, avoid mechanical forces such as shaking, moving, or otherwise handling the cells as much as possible. After 8 days of culture, the total cell yield should be similar to the cell input on Day 0 and overall approximately 40 × 10^6^ per mouse (2 femurs, 2 tibiae).Day 8:After the 8-day incubation, cells are harvested. After removing all the media, cells are collected by adding 0.25 mM EDTA (e.g., 2 mL/6-well) and incubated at 37 °C for no more than 3 min. Most cells should detach from the surface and the remaining attaching cells can be gently detached by flushing with the DC medium. The differentiation into DCs can be checked using flow cytometry.

Note 1: Depending on the application, DCs are most commonly harvested by flushing and collecting loosely adherent cells while discarding highly adherent cells which can contain higher proportions of macrophage-like cells [21]. But for the purpose of this investigation, we collected and compared all differentiated cells.

Note 2: A prolonged maintenance of cells in vitro may activate DCs and lead to increased cell death; hence, the primary cell culture should be used immediately after the 8-day differentiation period is completed.

### 3.5. Flow Cytometry Staining

Wash cells twice in PBS and resuspend at 5 × 10^5^ cells in 100 μL of live/dead Aqua plus Fc Block mix in 1× PBS per stain;Incubate cells for 20 min in the dark on ice;Wash cells with 100 μL 1× PBS and centrifuge at 400× *g* for 5 min at 4 °C;Re-suspend cells in 100 μL of antibody cocktail (MHC II-AF700, CD11c-APC, CD45R/B220-FITC, CD11b-PeCy7, CD24-eF450, and CLEC9A-BV605). Single-color controls with cells or compensation beads can also be prepared at this time;Incubate cells for 30 min in the dark on ice;Wash cells with 100 μL 1× PBS and centrifuge at 400× *g* for 5 min at 4 °C;(Optional): Re-suspend cells in 100 μL of 1% PFA and incubate in the dark at room temperature for 20 min. Wash cells with 100 μL 1× PBS and centrifuge at 400× *g* for 5 min;Re-suspend cells in 100 μL 1× PBS for flow cytometric analysis.

The cell phenotype was analyzed using FACSVantageSE DiVa on LSR II (BD Biosciences), and data analyzed using FlowJo™ v10.8.1 Software (BD Biosciences, Franklin Lakes, NJ, U.S.A.).

### 3.6. Statistical Analysis

Statistical analysis was performed using ratio paired *t*-tests and two-way ANOVA in GraphPad Prism 10 software. A *p*-value smaller than 0.05 was considered significant.

## 4. Expected Results

### 4.1. Total Cell Yields

The cell yields on Day 0 and Day 8 were compared between the two bone marrow isolation methods. To exclude any effects of inter-mouse variability, the isolation methods were set up to compare bone marrow isolation in the same mouse, i.e., one tibia and one femur were isolated using centrifugation and the other tibia and femur were isolated using the flushing method. The cell numbers isolated using both methods showed no significant difference on Day 0 (Figure 2A). Cells were seeded at the same density in 6-well plates as described and counted on Day 8. The centrifuged bone marrow culture generated significantly more cells after 8 days compared to the flushed-out bone marrow culture, with an average 1.32-fold ± 0.06 SD increase in total cells (Figure 2B). The isolation method had no effect on cell viability on Day 0 or Day 8 (Figure 2C,D).

### 4.2. Flow Cytometric Analysis of Surface Marker Expression

On Day 8, the primary culture cells were stained for specific surface markers and DC subsets were gated for phenotypic analysis (Figure 3).

The proportion of each DC subset was calculated based on their percentage of live single cells. The relative ratios of cDC1s, cDC2s, and pDCs remained constant between both isolation methods indicating no method-specific effects on the differentiation potential and phenotype of DCs (Figure 4A). This was also the case when the FSC/SSC gating was further sub-divided to individually assess smaller and larger cell subsets (Appendix A). The 1.32-fold increase in absolute cell counts together with unchanged DC proportions yielded moderately higher individual DC subset numbers in the centrifugation-isolated bone marrow culture (Figure 4B).

## 5. Conclusions

cDC1s play a crucial role in tumoral and viral immunity through the specialized cross-presentation pathway. To study cross-presentation, an efficient method is required for the generation of cDC1s. Previously described methods for murine bone marrow isolation have mainly focused on the manual flushing method. During isolation, the bone marrow is subjected to variable mechanical forces, at times being flushed out as small cell aggregates or an intact plug of bone marrow requiring rigorous agitation or pipetting to process it into a single-cell suspension. The flushing process itself applies mechanical forces to bone marrow cells which may affect their cell viability and surface protein expression, potentially resulting in the alteration of cell yield and differentiation capacity. The extended processing time of cells via flushing when a high number of bones are prepared, may also impair the primary culture outcomes. The number of bone marrow cells isolated using centrifugation was comparable to that using the conventional flushing method. After 8 days of culturing, the centrifugation-isolated bone marrow culture yielded a higher number of cells in total, and both methods showed similar proportions of DC populations. Overall, this resulted in a higher overall DC yield in the centrifugation-isolated bone marrow culture without phenotypic changes. Furthermore, in assays where differentiated DCs were exposed to heat-inactivated bacteria and MHC I peptides isolated using previously established protocols [22,23], cultures from bone marrow isolated via centrifugation successfully cross-presented bacterial peptide ligands (data not shown), indicating the efficient antigen cross-presentation capacity of cDC1s generated using the centrifugation protocol.

The higher cell yield of Day 8 cultures from centrifugation-isolated bone marrow may be explained by the different ratios of isolated DC precursors on Day 0. DC progenitors are not distributed evenly in the bone marrow [24]. The Flt3 receptor is mostly expressed by common lymphoid progenitors and CD16/32^int^ common myeloid progenitors. Common lymphoid progenitors have been shown to reside close to the endosteal surface of the bone marrow [24]. It is noted that centrifugation yields white bone, indicating the complete isolation of central and endosteal bone marrow while flushing has a less consistent outcome and can sometimes result in visible, hard-to-dislodge red residues of bone marrow even after several flushing cycles. This might possibly explain the overall better Day 8 yields of the Flt3L-driven murine bone marrow cultures isolated using centrifugation.

The 15 s centrifugation is considerably more time-efficient compared to the manual flushing approach. The duration of isolating bone marrow from four leg bones by flushing is estimated to be approximately 15–25 min, depending on the investigator. The handling time could be substantially reduced to approximately 10–15 min overall using centrifugation, with an even more substantially reduced workload when more bones are processed. In our hands, bone marrow progenitors can be highly sensitive to extended processing times; hence, a procedure that minimizes the time from bone isolation to in vitro culture is strongly recommended. It is worth mentioning that the centrifugation method is more standardized and operator-independent compared to the flushing procedure, allowing more consistent isolation of bone marrow cells and generation of DCs across researchers.

Further preliminary testing has shown that the centrifugation method additionally brings the benefit of being able to use the smaller humerus and ulna bones which can be easily processed via centrifugation but are too challenging to handle via the conventional flushing method. Thus, the centrifugation method closely aligns with the 3Rs of the replacement, reduction, and refinement principles of animal experimentation.

To conclude, the generation of Flt3L-driven murine bone-marrow-derived DCs is a long-standing practice in immunology with the isolation of bone marrow being the first pre-requisite to a successful in vitro culture. Here, we have described a step-by-step process to prepare single-cell suspensions from murine bone marrow with a rapid, consistent, and efficient centrifugal isolation method that yields a higher number Flt3L-driven functionally and phenotypically differentiated DCs. Overall, the centrifugation method is a faster alternative to the conventional bone marrow isolation method. This has relevance, particularly for experiments where high numbers of precursors are required.

## Figures and Tables

**Figure 1 mps-07-00020-f001:**
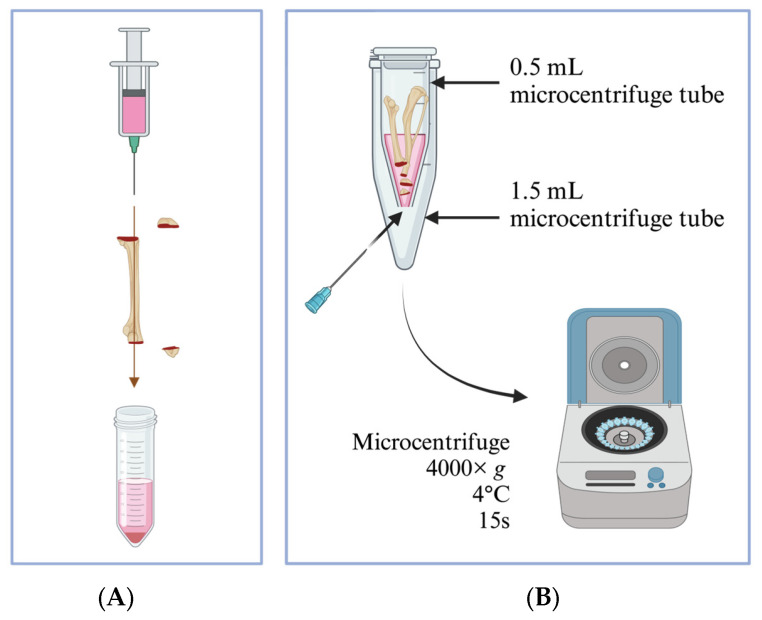
Isolation methods for murine bone marrow. (**A**) The flushing method. Bone marrow is flushed out from the bone cavity using a 21-gauge needle attached to a syringe filled with complete DC medium. (**B**) After cutting open the bone, place up to 2 bones and the epiphyses (cut sides facing downward) into a 0.5 mL microcentrifuge tube with a hole pierced at the bottom. Place the 0.5 mL tube containing bones into a 1.5 mL microcentrifuge tube, add 100 μL of ice-cold DC medium into the 0.5 mL tube, and close the lid. Centrifuge the set of tubes at 4000× *g* for 15 s at 4 °C. Observe for any remaining bone marrow (red color) in the bone cavity; bone appears white when completely void of bone marrow.

**Figure 2 mps-07-00020-f002:**
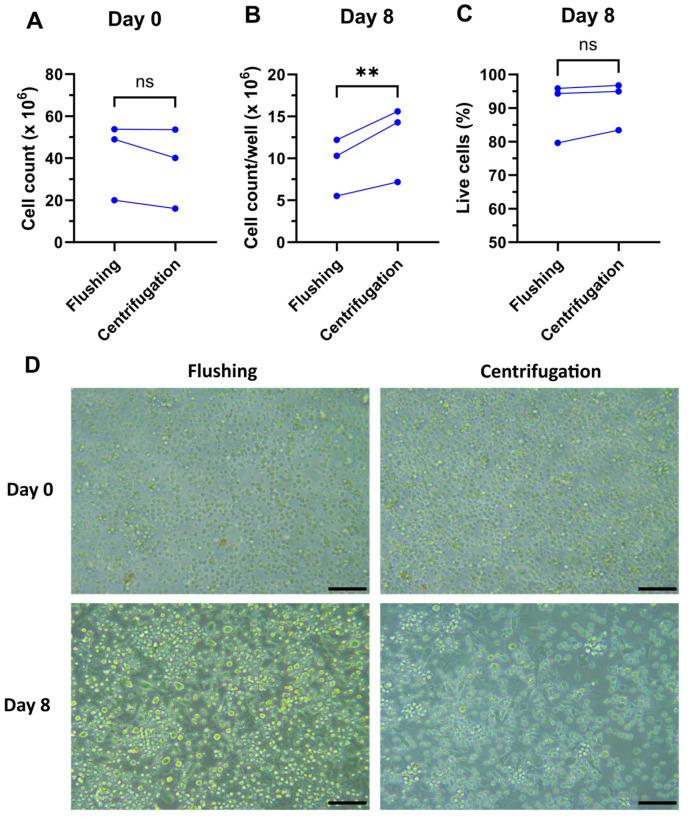
Total cell counts and viability of bone marrow and differentiated DC. (**A**) Number of total live cells isolated per mouse using flushing and centrifugation methods, extrapolated from 1 femur + 1 tibia. (**B**) Number of cells per well after 8 days of incubation. (**C**) Viability on Day 8 measured by flow cytometry. (**D**) Representative light microscopy images of different stages of cell culture. Scale bars: 50 µm. n = 3 independent experiments; ns: not significant; **: *p* < 0.01, ratio paired *t*-test.

**Figure 3 mps-07-00020-f003:**
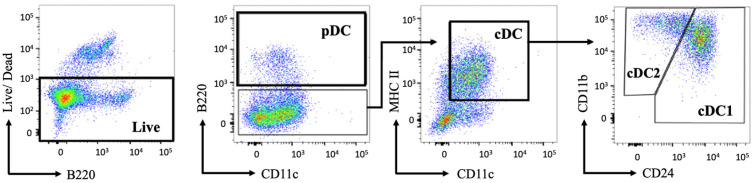
The gating strategy of DC subsets differentiated with Flt3L. After 8 days of differentiation, cells were stained with directly conjugated antibodies and analyzed using flow cytometry. After the pre-gating of single live cells, pDCs were gated as B220^+^, CD11c^+^, and MHCII^+^ (only B220 and CD11c gating is shown). cDCs were identified as double-positive for MHC II and CD11c. Subsequentially, cDC subsets were further identified by CD11b and CD24. cDC1s were gated as CD24^high^ and CD11b^int^. cDC2 were identified by the differential expression of CD11b^high^ CD24^int^. Representative gating of centrifugation-isolated murine primary bone marrow culture is shown; n = 3 independent experiments.

**Figure 4 mps-07-00020-f004:**
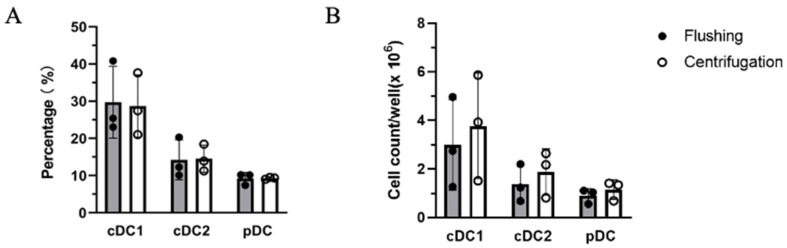
Comparison of DC differentiation on Day 8 using flushing and centrifugation bone marrow isolation methods. (**A**) Subset composition of cDC1s, cDC2s, and pDCs in live cell gating in flow cytometric analysis. (**B**) Cell number of cDC1s, cDC2s, and pDCs per well based on total live cell count and DC proportions. n = 3 independent experiments; data are presented as the mean ± SD.

**Table 1 mps-07-00020-t001:** Plasticware.

Material	Source	Identifier
Pipette filter tips: 10 µL 20 µL	Neptune Scientific San Diego, CA, U.S.A.	Cat# BT10 Cat# BT20
Stripettes: 5 mL	Corning Costar New York, NY, U.S.A.	Cat# CLS4487
Conical centrifuge tubes: 50 mL	Greiner Bio-One Singapore, Singapore	Cat# 227270
Microcentrifuge tubes: 0.5 mL 1.5 mL	Eppendorf Melbourne, Australia	Cat# 0030123603 Cat# 0030123611
Falcon 70 µm cell strainer	Corning Costar New York, NY, U.S.A.	Cat # 352350
Falcon 6-well plate-bottom plate	Corning Costar New York, NY, U.S.A.	Cat# 38016

**Table 2 mps-07-00020-t002:** Cell culture reagents.

Reagents	Composition	Source	Identifier
RF2	RPMI 1640 medium	Thermo Fisher Scientific Waltham, MA, U.S.A.	Cat# 21870-076
	2% *v*/*v* fetal calf serum (FCS)	Thermo Fisher Scientific Waltham, MA, U.S.A.	Cat# A3161001
Complete murine DC medium (308 ± 2 osmol/L)	RPMI 1640	Thermo Fisher Scientific Waltham, MA, U.S.A.	Cat# 21870-076
	10% *v*/*v* heat-inactivated FCS	Thermo Fisher Scientific Waltham, MA, U.S.A.	Cat# A3161001
	1% *v*/*v* GlutaMax	Thermo Fisher Scientific Waltham, MA, U.S.A.	Cat# 35050-061
	1% *v*/*v* HEPES	Thermo Fisher Scientific Waltham, MA, U.S.A.	Cat# 15630-080
	1% *v*/*v* penicillin/streptomycin	Thermo Fisher Scientific Waltham, MA, U.S.A.	Cat# 15140-122
	100 µM β-Mercaptoethanol	Sigma-Aldrich St. Louis, MO, U.S.A.	Cat# 60-24-2
2.5 mM EDTA	2.5 mM EDTA	N/A	N/A
1× PBS	1× PBS	N/A	N/A
Erythrocyte lysis buffer	8.3 g/L ammonium chloride in 0.01 M Tris-HCl buffer.	Sigma-Aldrich St. Louis, U.S.A.	Cat# R7757
In vivo MAb recombinant Flt3L-Ig (hum/hum)	N/A	Bio X Cell Lebanon, PA, U.S.A.	Cat# BE0098

**Table 3 mps-07-00020-t003:** Reagents for flowcytometry staining.

Reagents and Antibodies	Source	Identifier	Working Dilution (in 100 μL)
Live/Dead Fixable Aqua Dead cell Stain Kit	Thermo Fisher Scientific Waltham, MA, U.S.A.	Cat# 34957	1:500
Rat anti-mouse CD16/CD32 (mouse BD Fc block, clone 2.4G2)	BD Biosciences Franklin Lakes, NJ, U.S.A.	Cat# 553142	1:400
Rat anti-mouse/human CD45R (B220, clone RA-6B2)	Thermo Fisher Scientific Waltham, MA, U.S.A.	Cat# 11-0452-85	1:300
Rat anti-mouse MHC class II (clone M5/114.15.2)	Thermo Fisher Scientific Waltham, MA, U.S.A.	Cat# 56-5321-82	1:300
Hamster anti-mouse CD11c (clone HL3)	BD Biosciences Franklin Lakes, NJ, U.S.A.	Cat# 550261	1:300
Rat anti-mouse CD24 (clone M1/69)	Thermo Fisher Scientific Waltham, MA, U.S.A.	Cat# 48-0242-82	1:300
Rat anti-CD11b (clone M1/70)	BD Biosciences Franklin Lakes, NJ, U.S.A.	Cat# 552850	1:200
Rat anti-mouse CD370 (Clec9A, clone 10B4)	BD Biosciences Franklin Lakes, NJ, U.S.A.	Cat# 744511	1:100
Compensation beads	Thermo Fisher Scientific Waltham, MA, U.S.A.	Cat# 01-3333-42	1:20
PFA (Paraformaldehyde)	ProSciTech Kirwan, Australia	C004	1% in 1× PBS

## Data Availability

Data will be made available upon request.

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
