# Peer review of "A Simple and Rapid Protocol for the Isolation of Murine Bone Marrow Suitable for the Differentiation of Dendritic Cells"

_mps, 2024, doi:10.3390/mps7020020_

Round 1

Reviewer 1 Report

Comments and Suggestions for Authors

The manuscript entitled “A Simple and Rapid Protocol for the Isolation of Murine Bone Marrow Suitable for the Differentiation of Dendritic Cells”, by Runqiu Song and co-authors, describe an accelerated bone marrow isolation method using centrifugation, suitable for the differentiation of Flt3L-driven dendritic cells. Results suggest that the optimized centrifugation-based preparation is more time-efficient and yields equivalent numbers of DCs without changes to the phenotype, when compared to the conventional flushing method.

The protocol is simple and clearly written but the overall presentation can be improved in some aspects. Its usefulness is limited to a specific user niche.

Here are some comments:

-          To demonstrate the efficiency of the new method, the authors focused their attention on cells yield and viability, neglecting the functionality of DCs obtained with the innovative method. A test to evaluate the DC’s antigen uptake and presentation capacity should be shown.

-          Sample size is not defined: how many times did the authors replicate the experiments? How many mice per experiments? Moreover, in terms of yield, is 1 mouse compared to 1 mouse or is 1 tibia + 1 femur compared to 1 tibia + 1 femur?

-          Mouse strain and age, used for the experiments, is not defined. Is it irrelevant?

-          Paragraph 3 “Procedure”. It is not specified when, during procedure, the bones are transferred under a sterile hood.

-          Line 122. What type of scissors are used to cut bone heads? Since this is a protocol, it should be specified.

-          Line 125. Please, specify the volume of medium used to flush out the marrow from each bone.

-          Line 194. Here the authors specify that the DCs yield obtained after 8 days of culture corresponds to 40x10^6 cells per mouse; however, since neither the strain nor the age of the animals used in the experiments are specified, in my opinion it would be more correct to normalize this data with respect to the initial number of cells in culture.

-          - Both analyzed methods are characterized by extensive manipulation. Could bone centrifugation have a negative impact on microbial contamination compared to marrow flushing? Discuss.

-          Considering references 16 and 17, the authors should justify in more depth the usefulness of a further publication describing a protocol for mouse bone marrow extraction.

-          Line 30. Term “lymphocytes” looks like a mistake: should it be replaced with PBMCs?

Reviewer 2 Report

Comments and Suggestions for Authors

In this study, Song et al. conducted a study to assess the efficacy of an accelerated bone marrow isolation method using centrifugation for the differentiation of FMS-like tyrosine kinase 3 ligand-driven dendritic cells. They compared this method to the conventional flushing method. The results showed that the centrifugation-based isolation method yielded similar bone marrow cell numbers on day 0 but increased cell numbers by day 8, resulting in similar proportions of dendritic cell subsets and a higher number of type 1 conventional dendritic cells (cDC1) from the culture compared to the conventional method. 

The following suggestions could enhance the manuscript:

1.     It would enhance the comprehensiveness of the study if the authors could provide details such as the list of antibodies utilized for flow cytometry staining, the staining procedure, the specifications of the flow cytometer employed, and the software utilized for data analysis.

2.     If viable, it would be beneficial to include information regarding the cell viability observed in Figure 2A, as this could provide valuable insights into the overall health and functionality of the cells throughout the experiment.

3.     Additionally, it would be insightful to know if there were any microscopic comparisons conducted on the cells to assess their morphological similarity. Such observations could contribute to a more comprehensive understanding of the experimental outcomes.

Round 2

Reviewer 1 Report

Comments and Suggestions for Authors

I thank the authors for taking my suggestions into consideration and for editing the manuscript as indicated

Reviewer 2 Report

Comments and Suggestions for Authors

The manuscript is suitable for publication in its present form